# Pebax-Based Composite Membranes with High Transport Properties Enhanced by ZIF-8 for CO_2_ Separation

**DOI:** 10.3390/membranes12090836

**Published:** 2022-08-27

**Authors:** Tarik Eljaddi, Julien Bouillon, Denis Roizard, Laurent Lebrun

**Affiliations:** 1UNIROUEN, CNRS, PBS, Normandie Université, 76000 Rouen, France; 2Laboratoire Réactions et Génie des Procédés, CNRS, Université de Lorraine, 54000 Nancy, France

**Keywords:** mixed matrix membrane, Pebax, PEG, ZIF-8, CO_2_ separation

## Abstract

A series of mixed matrix membranes containing poly (ether-block-amide) Pebax 1657 as matrix and polyethylene glycol (PEG) and Zeolitic Imidazolate Framework-8 (ZIF-8) as additives, were prepared and tested for CO_2_ separation. The membranes were prepared by solvent evaporation method and were characterized by TGA, DSC, SEM, and gas permeation measurements. The effects of PEG and its molecular weight, and the percentage of ZIF-8 into Pebax matrix were investigated. The results showed that the addition of PEG to Pebax/ZIF-8 blends avoid the agglomeration of ZIF-8 particles. A synergic effect between PEG and ZIF was particularly observed for high ZIF-8 content, because the initial permeability of pristine Pebax was multiplied by three (from 54 to 161 Barrers) while keeping the CO_2_ selectivity (α_CO_2_/N_2__ = 61, α_CO_2_/CH_4__ = 12 and α_CO_2_/O_2__ = 23). Finally, the mechanism of CO_2_ transport is essentially governed by the solubility of CO_2_ into the membranes. Therefore, this new Pebax/PEG/ZIF-8 system seems to be a promising approach to develop new selective membranes for CO_2_ with high permeability.

## 1. Introduction

Carbone dioxide (CO_2_) is one of the main contributors to global warming and climate change. The rejection of CO_2_ in atmosphere increases the temperature of Earth and makes the life of several species in critical situation. Thus, the capture of CO_2_ becomes one of the big challenges of our century [1]. In addition, CO_2_ in biogas and natural gas must be removed because it reduces the calorific capacity of gases and the acidic property of CO_2_ degrades the gas streams by corrosion phenomenon [2].

Several conventional techniques are used to solve these problems, such as amine absorption, but these technologies suffer of their high cost (high capital and operating investment) and high carbon footprint. On contrary, the membrane technology is a more ecofriendly alternative, with much lower footprint, low operational expenditures and moderate capital costs, does not involve phase changes or chemical additives, and is modular and easy to scale up [3,4,5]. However, the membrane technologies also have some drawbacks, such as limited selectivity compared to chemical separation processes, decreasing performances with time due to aging or fouling effects, and difficulties to operate at higher temperature than 150 °C with polymeric membranes. For these reasons, researchers works on the development of new generations of membranes using pure inorganic materials such as graphene [6], carbon nanotubes [7] and metal organic frameworks [8]. Mixed matrix membranes (MMM) are also developed, that consist to introduce inorganic materials into polymeric matrixes in order to improve the performance of the pristine membranes [9,10,11,12,13,14,15].

Polymeric materials are the most used to prepare membranes because they are less expensive and easier to scale up than inorganic materials. The most used polymers for gas separation are cellulose acetate, polysulfone, polyimide, polycarbonate, and Pebax. These last are block-copolymers that contain polyamide PA-6 or PA-12 hard segments that gives strong mechanical resistance to the material, and poly(ethylene oxide) (PEO) or poly(ethylene glycol) (PEG) amorphous segment that are able to induct strong interactions with CO_2_ [5,16]. Consequently, Pebax copolymers are widely studied for CO_2_ separation. Several grades are available for CO_2_ separation depending on the percentage and the nature of the hard and flexible segments [17,18].

Inorganic membranes present higher performances than polymeric membranes. So, to reach high separation performances, new generation of membranes were developed by mixing inorganic frameworks and organic polymers. Several studies on these mixed matrix membranes (MMMs) are described in the literature [17]. One of the most attractive nanofillers used in MMMs are the Zeolite imidazolate frameworks (ZIFs). ZIFs are built with tetrahedral metal ions (Zn) bridged by imidazolate groups. They show a constant porosity and relatively high thermal and chemical resistances. For example, ZIF-8 have large pore cavities (1.16 nm) that are connected with a small pore aperture (0.34 nm). The small pore size makes ZIF-8 very interesting for CO_2_ separation by molecule sieving mechanism because the kinetic diameter of CO_2_ is 0.33 nm. Many papers highlight the increase in the permeability as well the selectivity of MMMs with ZIF-8 loading. Askari and Chung prepared MMMs by mixing ZIF-8 with 4,4′-hexafluoroisopropylidene diphthalic anhydride (6FDA)-based poly(imide) [19]. The CO_2_ permeability increased from 256 Barrers without ZIF-8 to 779 Barrers with 40% of ZIF-8 loading without decrease of the CO_2_/CH_4_ selectivity (α_CO_2_/CH_4__ ≈ 20). Nafisi and Hägg [20] developed membranes by mixing ZIF-8 (5% to 35%) into Pebax 2533 matrix. The CO_2_ permeability increased from 351 to 1287 Barrers without change in separation factor. Jomekian et al. [21] prepared composite membranes using ZIF-8/Pebax 1657 as dense layer supported by poly(ether sulfone), where the permeability for CO_2_ was doubled with constant selectivity. Pebax was also used in our study. PEO groups contained in Pebax can increase the CO_2_ permeability by increasing the CO_2_ solubility due to the quadrupole-dipole interactions between polar ether groups and CO_2_, and can increase the diffusion because PEO polymers are highly flexible [18]. Furthermore, Pebax contains also block polyamide as hard segments that provide mechanical stability to the membrane and inhibit crystallization of PEO. However, the problem of aggregate formation of inorganic particles remains a real challenge. To avoid this problem, several strategies are used, such as heat treatments [22,23], sonication [24] or the additions of others MOFs or polymers [25]. Thus, polymers containing polar groups such as PEG can be used as surfactant to reduce the aggregates formation of inorganic particles [26,27]. The main objective of the study was to investigate the effect of the addition of PEG in Pebax/ZIF-8 MMMs on the CO_2_/N_2_ and CO_2_/CH_4_ separations.

## 2. Material and Methods

### 2.1. Chemicals

Pebax 1657 containing 60 wt% block polyether and 40 wt% block polyamides was graciously provided by Arkema (Serquigny—France). Basolite Z1200 (ZIF-8 MOF) (M = 227.6 g mol^−1^) was generously provided by BASF (Ludwigshafen—Germany). Polyethylene glycol PEG 300 (300 g mol^−1^), PEG 600 (570–630 g mol^−1^) and PEG 1500 (1400 g mol^−1^) were supplied by Merck (Sigma-Aldrich, St. Louis, MO, USA), Germany. Absolute ethanol was purchased from VWR, USA. Ultrapure gases used for the permeation tests were supplied by Messer France. All gases and molecules were used as received.

### 2.2. Membrane Preparation

Neat Pebax and Pebax-based hybrid membranes were prepared via solution casting method. Pebax 5 wt% was first dissolved in ethanol/water (70:30 *v*/*v*) solution under stirring for 4 h at 90 °C. A specific protocol was used to avoid ZIF-8 aggregates. Firstly, ZIF-8 particles (7, 13, 23 wt% ZIF-8/Pebax) were dispersed under stirring in ethanol/water solution (70:30 *v*/*v*) for 15 min. After sonication for 15 to 20 min, PEG (around 33 wt% according to Car et al. [28] was added and stirred for 15 min, and then the Pebax solution. The Pebax/PEG/ZIF ratios for each prepared membrane are given Table 1. The suspensions were stirred carefully to remove gas bubbles before casting on Petri dishes. After drying for one night, the obtained films were carefully removed of the support and dried in an oven at 60 °C for 12 h. Finally, the films (membranes) were placed in desiccator under vacuum to remove residual solvent. The thickness of the dry membranes was in the range of 80 µm micrometers with an average deviation of 1 µm.

### 2.3. Thermal Characterization (TGA and DSC)

Thermal properties of membrane samples were characterized by differential scanning calorimetry (DSC) with a DSC 214 Polyma from Netzsch, Germany. The temperature and energy calibration were performed with indium standard (*T_m_* = 156.6 °C, $\Delta H_{m}=28.66$ J·g^−1^). The measurements were carried out on 8–10 mg membrane samples placed in pierced an aluminum crucible from −120 to 250 °C at 20 °C min^−1^. All measurements were conducted under nitrogen purge gas.

The degree of crystallinity ***X_C_*** of the polymers in the membranes were calculated from DSC thermograms using the following equation [29]:(1)$$X_{C}=\frac{\Delta H_{m}}{\Delta H_{m}^{{}^{\circ}}}\times 100$$
where $\Delta H_{m}$ (J·g^−1^) is the melting enthalpy calculated by integrating the area of endothermic melting peaks; $\Delta H_{m}^{{}^{\circ}}$ is the theoretical melting enthalpy considering the polymer 100% crystalline (values obtained from literature: PEO: $\Delta H_{m}^{{}^{\circ}}$ = 166.4 J·g^−1^, PA: $\Delta H_{m}^{{}^{\circ}}$ = 230.0 J·g^−1^ [29].

In addition, thermogravimetric measurements were performed using TG 209 Libra from Netzsch, Germany on 8–10 mg membrane samples from 30 °C to 1000 °C at 20 °C min^−1^ under N_2_.

### 2.4. Scanning Electron Microscopy (SEM-EDX)

Surface and cross-section morphologies of the prepared membranes were observed by scanning electron microscopy (SEM) coupled with EDX analyzer with SEM EVO40EP from Zeiss, Germany. The cross-sections were obtained by fracturing the samples in liquid nitrogen. All samples were first covered with a thin layer of gold before study.

### 2.5. Gas Permeation Measurements

CO_2_, O_2_, N_2_ and CH_4_ permeation measurements on the membranes were performed at 25 °C using a barometric method based on the time-lag determination [30,31,32]. The measurement cell containing the membrane was placed under high vacuum for 24 h, before to carry out the measurement by applying 3 bars of pure gas to the upstream side of the membrane. The quantity of transferred gas *Q* through the membrane vs. time *t* was monitored until the increase in pressure in the downstream side reached a constant value indicating the stationary state of the permeation process. The permeability coefficient *P* (expressed in Barrer unit, 1 Barrer = 10^−10^ cm^3^ (STP) cm cm^−2^ s^−1^ cm Hg^−1^) was calculated from the following equation:(2)$$P=\frac{J_{st}\cdot L}{\Delta p}=\frac{L\;}{\Delta p\cdot A}\times \frac{dQ}{dt}$$

$J_{st}$ is the stationary flow, *A* the active area is 11.34 cm^2^ and 8.04 cm^2^ for the cell used for CH_4_), L the thickness of membrane and Δp is the difference in pressure between feed (upstream) and permeate (downstream) sides of the membrane.

The diffusion coefficient *D* was calculated by the equation:(3)$$D=\frac{L^{2}}{6t_{L}}$$
where *t*_L_ is the time lag calculated from the diffusion curve by the extrapolation of the steady-state asymptote to the time axis.

Based on solution–diffusion transport model, the solubility *S* was calculated by the equation:(4)$$S=\frac{P}{D}$$

The selectivity coefficient α between gas 1 and gas 2 is defined as a ratio of the permeability of the more permeable gas to the less permeable:(5)$$\alpha_{1/2}=\frac{P_{gas\;1}}{P_{gas\;2}}$$

For each membrane, the permeation experiments were performed at least three times and the results were averaged.

## 3. Results and Discussion

### 3.1. Effect of PEG Molecular Weight on Pebax/PEG Blend Membranes

#### 3.1.1. Morphology of Membranes

The surfaces and cross-sections morphologies of Pebax 1657 and Pebax 1657/PEG blend membranes are shown in Figure 1. All membranes showed dense structures. In addition, the surface and cross-section of Pebax (P) membranes and Pebax/PEG blend (PP300, PP600) membranes containing low molecular weight PEG (300 to 600 g mol^−1^) showed homogeneous structures. However, in Pebax/PEG 1500 (PP1500) blend membrane containing higher molecular weight PEG, some micro-phase separations were observed on the surface and on the cross-section due to the presence of crystals of PEO. This presence is due to the highest crystallinity ratio of PEG that increases with the molecular weight [33]. This change in the membrane morphology could affect the gas transport properties.

#### 3.1.2. Thermal Analysis

Pebax 1657 copolymer has two blocks: the first one contains PEO groups (60 wt%) which gives flexibility to the polymer and the second one contains hard (rigid) PA segments (40 wt%). The addition of flexible PEG polymers to Pebax will change the composition of the membrane matrix, and consequently could affect its physical properties.

DSC thermograms of neat Pebax and Pebax/PEG blend membranes are given Figure 2 and the values associated to the thermal phenomena are reported Table 2. First, it can be noticed that only glass transition of PEO segments was detected [28]. The glass transition *T*g of neat membrane decreases with the addition of PEG 300 in membrane due to the plasticizing effect of the small flexible PEG chains. However, *T*g values increase according to the molecular weight of PEG: *T*g = −70, −61, −50 °C for PEG 300, 600, 1500 g mol^−1^, respectively. This change in *T*g is caused by the chains stiffening due to the increasing molecular weight of PEG.

In the same direction, neat Pebax showed two melting temperatures at 18 and 199 °C corresponding to the fusion of crystalline fraction of the PEO and PA blocks, respectively [34]. On Pebax/PEG blend membranes, the melting point of PEO segment increased with the molecular weight of PEG loaded in Pebax matrix. This result agrees with the crystallinity values that also increased with molecular weight of PEG. Consequently, the low crystallinity and the formation of free volumes is expected to improve the chain mobility which is favorable for gas diffusion.

The thermal stability of neat Pebax and Pebax/PEG blend membranes were investigated by TGA (Figure 2), and the corresponding decomposition temperatures, T1% and T5% at 1 and 5% weight loss, respectively, and temperatures of the derivative peak (DTG) are listed Table 2. The obtained results showed that the membranes should remain stable up to 200 °C whatever the PEG content. However, the presence of low molecular weight PEG 300 decreased the thermal stability.

#### 3.1.3. Gas Permeation Measurements

CO_2_, N_2_, O_2_ and CH_4_ permeation measurements were performed on Pebax/PEG blend membranes to study the influence of the molecular weight of PEG. The obtained permeability *P* and selectivity coefficients are gathered Figure 3 and the accuracy on *D*, *P* and *S* values is 7%, 7% and 14%, respectively. The permeability coefficient of CO_2_ was 52 Barrers for the neat Pebax and the selectivity coefficients CO_2_/N_2_ CO_2_/CH_4_, and CO_2_/O_2_, were 54, 16 and 21, respectively. These results agree with the values found in literature [15,35,36].

The addition of PEG 300 and PEG 600 (low molecular weight) to the Pebax matrix significantly increased the permeability of CO_2_ and CH_4_ and the selectivity CO_2_/N_2_ and CO_2_/O_2_. The effect was more marked for PEG 300, at about 50% more. The addition of PEG300 to Pebax matrix increased the CO_2_ permeability to 73 Barrers and the CO_2_ selectivity towards N_2_ and O_2_ (α_CO_2_/N_2__ = 71, α_CO_2_/O_2__ = 27) but slightly decreased the selectivity towards CH_4_ (α_CO_2_/CH_4__ = 12). These variations were caused by favorable interactions between ethylene oxide groups present in PEG (and Pebax) with CO_2_, because CO_2_ is a polar molecule. However, the addition of PEG 1500 decreased drastically the permeability and selectivity of gases and particularly CO_2_. Similar results were reported by Kargari et al. [16] and also they were found by Feng et al. [37], and were explained by the high crystallinity of PEG 1500 (***Xc*** PEO = 49% in the membrane) (see Figure 1 and Table 2). As known, the permeability of CO_2_ through dense membrane agrees with the solution-diffusion model. Then, the solubility of CO_2_ at 25 °C (temperature of the measurement) is higher in PEG 300 and PEG 600 amorphous phase (lowest ***Xc*** PEO) than in PEG 1500 (largest crystalline phase) (Table 2. The obtained solubility and diffusion coefficients on the different gases confirm this conclusion (Figure 4). CO_2_ was more soluble in membranes containing PEG 600, but its diffusion was slower. As a compromise solution, PEG 300 was selected for the next studies on the effect of ZIF-8 because it makes the membranes more selective to CO_2_.

### 3.2. Effect of ZIF-8 and PEG on Pebax Based Membranes

#### 3.2.1. Morphology

SEM images of surfaces and cross-section of the ZIF-8 and PEG on Pebax based membranes are showed in Figure 5. The morphology of Pebax ZIF-8 (7 wt%) membranes, without (P + 7% ZIF) and with PEG300 (PP + 7% ZIF), were similar but appeared rougher than neat Pebax. In addition, ZIF-8 particles can be clearly observed on the cross-sections and surfaces of the membranes containing larger amount of ZIF-8 (13 and 23 wt%). Higher than 23% ZIF-8 content gives defects in the membranes and decreases the CO_2_ selectivity. Moreover, few ZIF-8 agglomerates can be observed on the membrane containing 13% of ZIF-8 (P + 13% ZIF-8). This problem of aggregation is mentioned in several studies with different types of particles, such as TiO_2_, SiO_2_ and zeolite [38,39,40,41]. EDX analyses were used to evaluate the ZIF-8 dispersion and the influence of PEG polymers (Figure 6). The blue color highlights the dispersion of Zn elements contained in ZIF-8 particles. The EDX analyses clearly indicate an improvement of the ZIF-8 dispersion in Pebax matrix containing PEG (Figure 6B). Without PEG, agglomeration of ZIF-8 particles and large dark zones corresponding to poor ZIF-8 zones were observed (Figure 6A). Ghadimi et al. mentioned that PEG has a double role: first, it can cover the particles to avoid the formation of hydrogen bonds between solvent and particles; second, PEG gives favorable interactions with Pebax for the gas transport [39]. So, PEG polymers can get homogenous MMMs by increasing the dispersion of particles into polymer matrix.

#### 3.2.2. Thermal Analysis

Thermal properties of Pebax/ZIF-8 and Pebax/PEG/ZIF-8 systems were investigated to check the thermal stability of the membranes, to find relations between structure and transport properties and to highlight the role of PEG in the Pebax/ZIF-8 blends. The thermogram were given Figure 7 and Figure 8. In addition, the corresponding data were gathered in Table 3.

Pebax/ZIF-8 membranes containing 7 and 13 wt% of ZIF-8 were first studied and compared with neat Pebax. As mentioned before, neat Pebax thermogram showed two pics corresponding to the melting points of PEO (*T* = 18 °C) and PA (*T* = 198 °C) segments (Figure 7). The addition of ZIF-8 slightly increased the *T*_g_ of membranes (from −49 to −47 °C) but decreased the crystallinity degree of PEO and PA, due to an improvement of chains stiffness by interactions between ZIF-8 particles and Pebax [20,42,43]. The TGA analyses showed a slight influence of the ZIF-8 load with the decrease of the thermal stability of the membranes (Table 3). Similar results were found in literature [29]. It can be noticed that ZIF-8 particles were more stable than neat Pebax polymer.

The addition of PEG to avoid the agglomeration of ZIF-8 particles decreased the *T*_g_ of neat Pebax and Pebax/ZIF-8 membranes due to the plasticizing effect of PEG (Figure 8). The endothermic peak between −20 and 50 °C was attributed to melting point of PEO segment of Pebax and PEG polymer. For ZIF-8 7 wt% (PP + 7% ZIF) three peaks were observed and the crystallinity degree was higher (***Xc*** = 36%) than for the other ZIF-8 content. The presence of broad peaks or many separated peaks mean that the crystals in the membranes are non-homogeneous in size. For higher than ZIF-8 7 wt% content, the degree of crystallinity attributed to PEO segments was lower than in neat Pebax. On the other hand, they are an influence of PEG but no real influence of ZIF-8 on the crystallinity attributed to PA segments. Thus, there are obviously interactions between PEG and Pebax and between PEG and ZIF-8 depending on the ZIF-8 ratio, which can be favorable or non-favorable for the crystallization. The large increase in the crystallinity observed for Pebax/PEG/ZIF-8 7 %wt membrane could be favorable for gas separation.

The TGA measurements indicated that addition of PEG decreased the thermal stability of blend membranes, but the membranes were still stable up to 200 °C (Figure 8).

Consequently, the use of PEG to prepare the mixed membranes is favorable to improve the dispersion of ZIF-8 particles and influence the degree of crystallinity of the membrane.

#### 3.2.3. Gas Permeation Measurements

CO_2_, CH_4_, O_2_ and N_2_ gas permeation measurements were performed on Pebax/ZIF-8 and Pebax/PEG/ZIF-8 blend membranes and compared to neat Pebax (Figure 9) and the accuracy on *D*, *P* and *S* values is 7%, 7% and 14%, respectively.

The addition of ZIF-8 (7 and 13 wt%) in Pebax significantly increased the CO_2_ permeability, from 52 Barrer for pristine Pebax to 129 Barrer for Pebax containing ZIF-8 13 wt%. The selectivity for CO_2_ were increased with ZIF-8 7 wt% but decreased with higher ZIF-8 content. The effect was more marked for α_CO_2_/N_2__ than for α_CO_2_/O_2__ and α_CO_2_/CH_4__. These results can be explained by the high selective properties of ZIF-8 for CO_2_ separation from gases. ZIF-8 is highly selective to CO_2_ due to its molecular sieving properties [22]: the pore size of ZIF-8 is 3.4 Å [44] and kinetic diameters of CO_2_, CH_4_, O_2_ and N_2_ are 3.3, 3.8, 3.46 and 3.64 Å, respectively [45]. Moreover, ZIF-8 particles can also create additional free volume in polymer matrix leading to preferential diffusion pathways and so, an increase in the permeability [42]. For the highest ZIF-8 content (23 wt%), a strong increase in permeability is observed for all gases (except N_2_) due to the larger free volume. Similar observations are mentioned in literature [29]. It could be supposed that the CO_2_ molecules would diffuse more easily than other molecules in free volume along matrix/ZIF-8 interfacial zones due to its lowest diameter, but it is not the case. The interfacial zones seem to be too large to improve the selectivity between gases.

As previously seen in Section 3.1.3, the addition of PEG 300 to Pebax matrix increased the CO_2_ permeability and the selectivity α_CO_2_/N_2__ and α_CO_2_/O_2__ due to favorable interactions between polar CO_2_ molecules and ethylene oxide groups in PEG. The addition of PEG to Pebax/ZIF-8 blends slightly increased the CO_2_ permeability and selectivity whatever the gas. The strong increase in the permeability of the gases observed for 23 wt% ZIF-8 content was previously explained by the increase in the free volume in the membrane. It could be also supposed that PEG would limit this free volume by covering the ZIF-8 particles, but it was not the case. In addition, *P*_N_2__ and *P*_O_2__ were not influenced by PEG: e.g., for 13 wt% ZIF-8 *P*_N_2__ and *P*_O_2__ are 2.3 and 6.0 Barrer with PEG and 2.2 and 5.7 Barrer without PEG, respectively. The increase in *P*_CO_2__ and *P*_CH_4__ with PEG is only due to the favorable interactions between these gases and PEG as observed previously in the experiments without ZIF-8.

Finally, the addition of PEG and ZIF-8 to Pebax showed a positive effect for CO_2_ separation because the permeability and the selectivity were increased. The synergic effect between PEG and ZIF was particularly marked for high ZIF-8 content where the initial permeability of Pebax was multiplied by three (from 54 to 161 Barrers) while keeping the CO_2_ selectivity.

The solubility and diffusion coefficients were calculated (Figure 10) using the solution-diffusion model in which the permeability is the product of the solubility (*S*) by the diffusion (*D*) for the gas transport through dense membrane. Whatever the membrane, *P* values followed the decreasing order P_CO_2__ >> P_CH_4__ > P_O_2__ > P_N_2__ (Figure 9) while *D* values did not change much (except for PP containing ZIF-8 23 wt%) (Figure 10). On the other hand, *S* coefficients highly depended on the type of gas, with 10-times larger *S* values for CO_2_. It can be explained by the polar character of CO_2_ molecules that involve quadrupoles [16] interactions with ethylene oxide groups in membrane. Thus, the transport of CO_2_ through the membranes was more controlled by the solubility than the diffusion.

Other factors can influence the diffusivity and the solubility of molecules in dense membranes such as shape, size, and condensability of the diffusing molecules, and the polymer and additives. The largest *P*_CO_2__ values obtained for all membranes are almost explained by the strong interactions between CO_2_ molecules and the ethylene oxide groups in PEG and Pebax due to its highest polarizability, and by the smallest size of CO_2_ compared to others tested gases (N_2_, O_2_, CH_4_) [37,46].

### 3.3. Effect of Temperature on Permeation Properties

To study the effect of temperature on gas permeability additional measurements were performed at 30, 35 and 40 °C on Pebax, Pebax/PEG, Pebax/ZIF-8 and Pebax/PEG/ZIF-8 membranes (PEG 300, ZIF-8 13% wt) (Figure 11 and Figure 12) and the accuracy on *P* values is 7%.

For all membranes, the permeability coefficients (Figure 11) increased with temperature while the selectivity (Figure 12) decreased. Pebax/PEG/ZIF-8 membrane showed the highest permeability values whatever the temperature and the gas. These results were in agreement with those obtained with UiO-66-Pebax and UiO-66-NH_2_-Pebax mixed matrix membranes system [47]. The increase in permeability by temperature was due to the total melting of crystalline fraction of the PEO (from Pebax and PEG), the melting of the large crystalline fraction observed for Pebax/PEG/ZIF-8 blend (between −20 and 50 °C) and the creation of more free volumes caused by the thermal motions of the polymer chains in the amorphous regions [48]. A decrease in selectivity was also observed with temperature with a similar tendency whatever the membrane. At 40 °C, there is no more influence of the ZIF-8 load and PEG because α_CO_2_/N_2__ and α_CO_2_/O_2__ have the same values whatever the membrane. With the increase in temperature, α_CO_2_/CH_4__ values also decreased but were less affected than the other selectivity. In conclusion, with the increase in temperature, the permeability of membranes is increased but the selectivity for CO_2_ decreased relatively. 

Furthermore, the effect of temperature on the gas transport can be expressed by the Arrhenius equation [49]:(6)$$PP_{0}.\exp\left(-\frac{E_{a}}{RT}\right)$$
where *P* is the permeability coefficient *P*_0_ a pre-exponential factor, R the gas constant (kJ mol^−1^ K), *T* the temperature (expressed in K) and $E_{a}$ is the activation energy in kJ mol^−1^.

The graphical representation of Ln *P* = f(1000/T) allowed to determine the activation energy of the tested gas. The obtained values are gathered Figure 13. All activation energy values were positive that confirms the increase in permeability with the temperature. The obtained values for pristine Pebax were similar to literature values [50].

CO_2_ activation energy value for neat Pebax (19 kJ mol^−1^) highly decreased in presence of PEG (8 kJ mol^−1^). This result was previously explained by the favorable interactions between polar CO_2_ molecules and ethylene oxide groups in PEG. The CO_2_ permeation mechanism need less energy to be performed with PEG that increased the CO_2_ permeability and selectivity. *E*_a_ values also decreased when ZIF-8 was added to Pebax and Pebax/PEG (13 and 14 kJ mol^−1^, respectively). The highest *E*_a_ values for O_2_, N_2_, CH_4_ were explained by their lowest solubility in the matrix membranes.

## 4. Conclusions

In this study, a series of Pebax composites membranes were prepared with different additives (PEG, ZIF-8) and tested for CO_2_ separation from air and CH_4_. ZIF-8 was used as sieving particles to improve the CO_2_ selectivity. PEG was used to improve the dispersion of ZIF-8 and to increase the CO_2_ solubility in the membranes.

The experiment results showed that the addition of low molecular weight PEG in Pebax significantly increased the permeability of CO_2_ and CH_4_ and the selectivity CO_2_/N_2_. These results were explained by favorable polar interactions between CO_2_ and ethylene oxide groups in PEG. However, the addition of high molecular weight (PEG 1500) drastically decreased the permeability and selectivity for CO_2_ due to its high crystallinity. In conclusion, low molecular weight of PEG is recommended to improve the performance of the Pebax membranes.

As expected, the addition of low ZIF-8 contents in Pebax membranes significantly increased the CO_2_ selectivity by a molecular sieving mechanism. The CO_2_ permeability was also improved due to the increase in the free volume in the matrix/ZIF-8 interfacial zones in polymer matrix. However, the effect of ZIF-8 particles was less marked than with PEG. Moreover, for higher ZIF-8 content (23 wt%), the increase in the free volume for all gases lead to a decrease of CO_2_ selectivity.

The incorporation of PEG into Pebax/ZIF-8 membrane clearly avoids the agglomeration of ZIF-8 particles. It also showed a positive effect for the CO_2_ permeability because the mechanism of CO_2_ transport in the membrane was essentially governed by the solubility. This effect was particularly marked for high ZIF-8 content where the initial permeability of Pebax was multiplied by three. However, the selectivity towards CO_2_ were less than those obtained without ZIF-8. There is a compromise to find between the improvements of permeability and selectivity. For example, the use of a highest selective CO_2_ Pebax/PEG/ZIF-8 as thin layer coated on a porous support could increase both permeability and selectivity.

In conclusion, the new Pebax/PEG/ZIF-8 system can be a promising approach to develop selective membranes for CO_2_ with a high permeability.

## Figures and Tables

**Figure 1 membranes-12-00836-f001:**
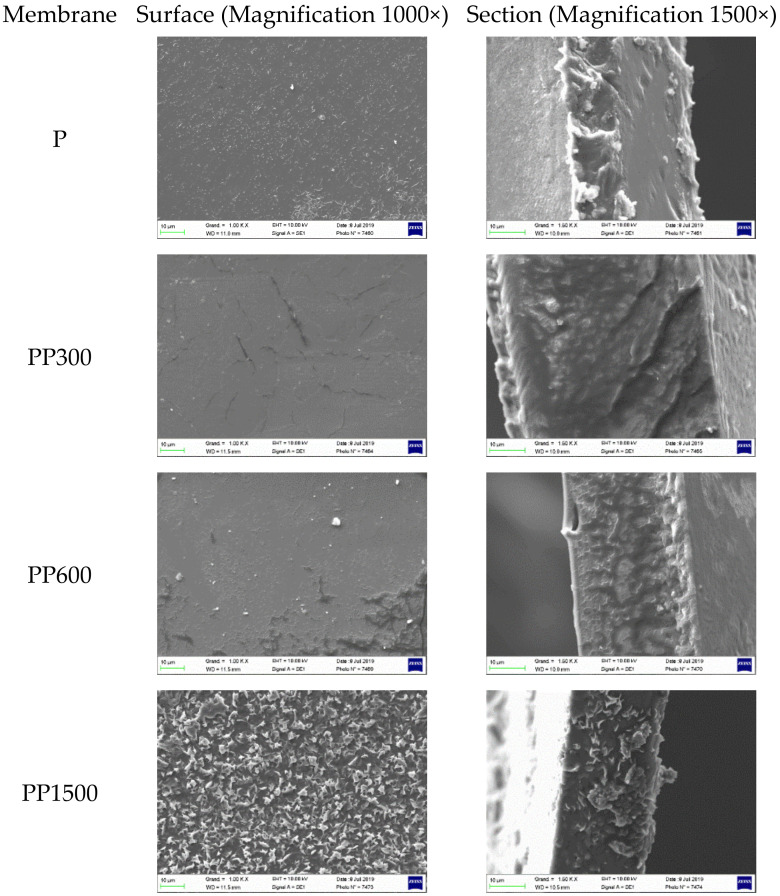
Surface and cross-section morphology of neat Pebax and Pebax/PEG (67/33 wt%) membranes.

**Figure 2 membranes-12-00836-f002:**
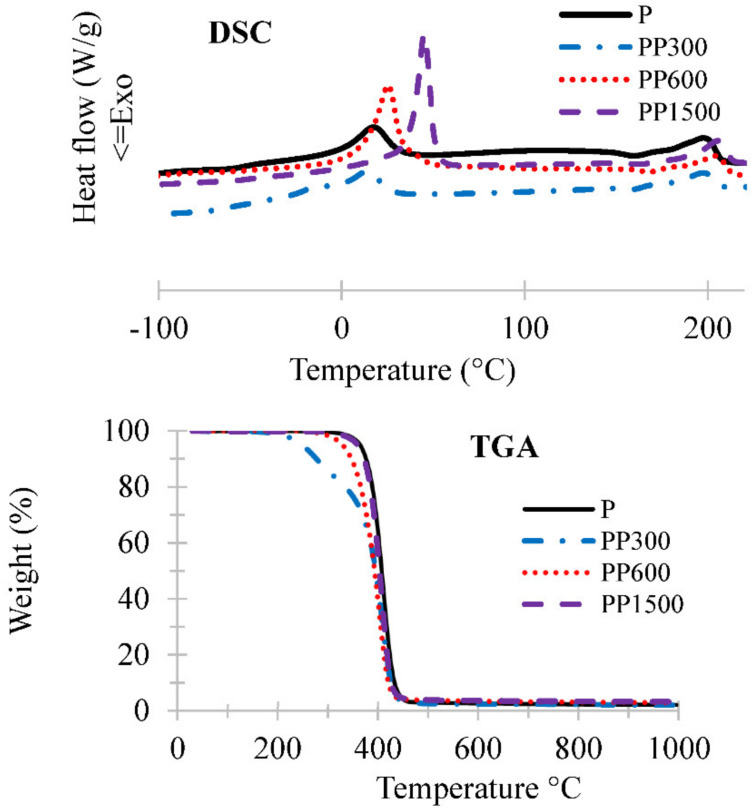
TGA and DSC curves of neat Pebax and Pebax/PEG membrane.

**Figure 3 membranes-12-00836-f003:**
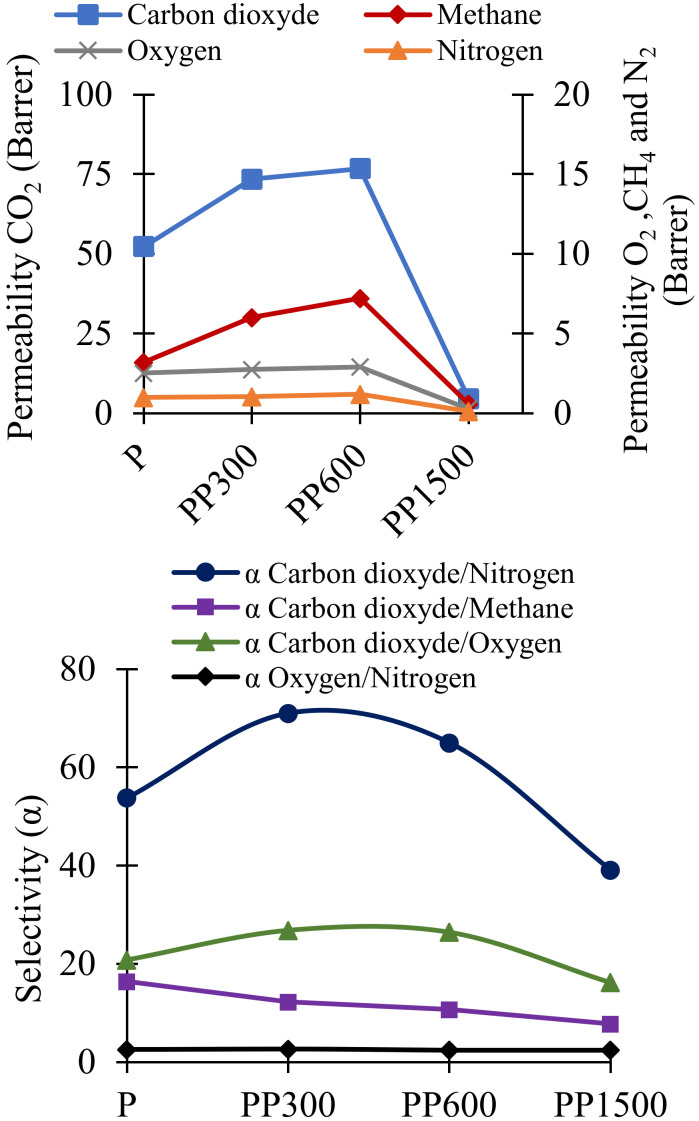
Permeability and selectivity coefficients of CO_2_, CH_4_, N_2_, and O_2_ for neat Pebax and Pebax/PEG membranes.

**Figure 4 membranes-12-00836-f004:**
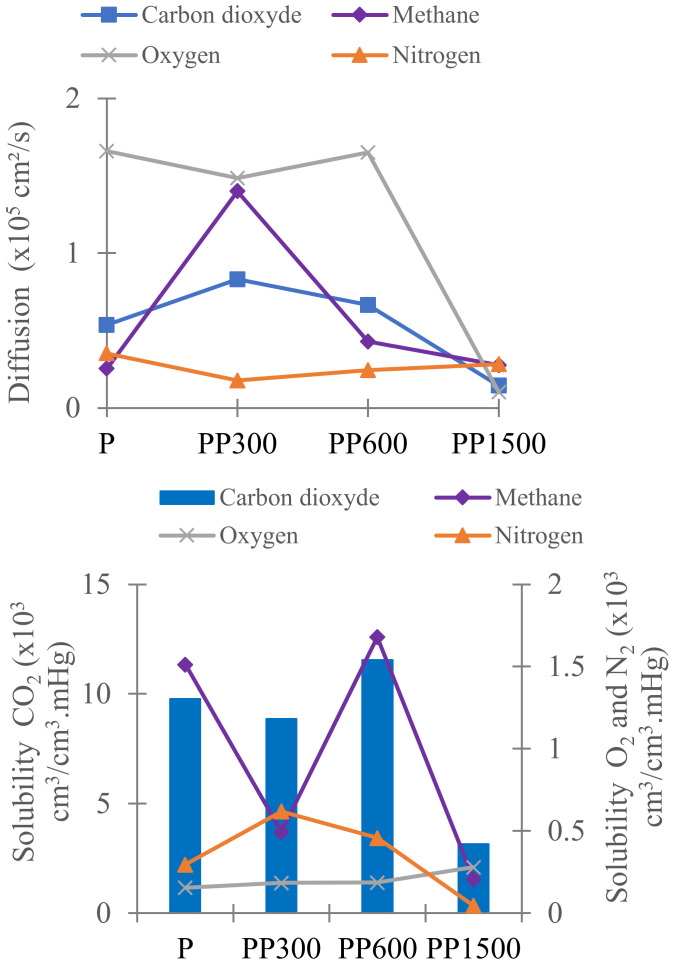
Solubility and diffusion coefficients of CO_2_, CH_4_, N_2_, and O_2_ of neat Pebax and Pebax/PEG membranes.

**Figure 5 membranes-12-00836-f005:**
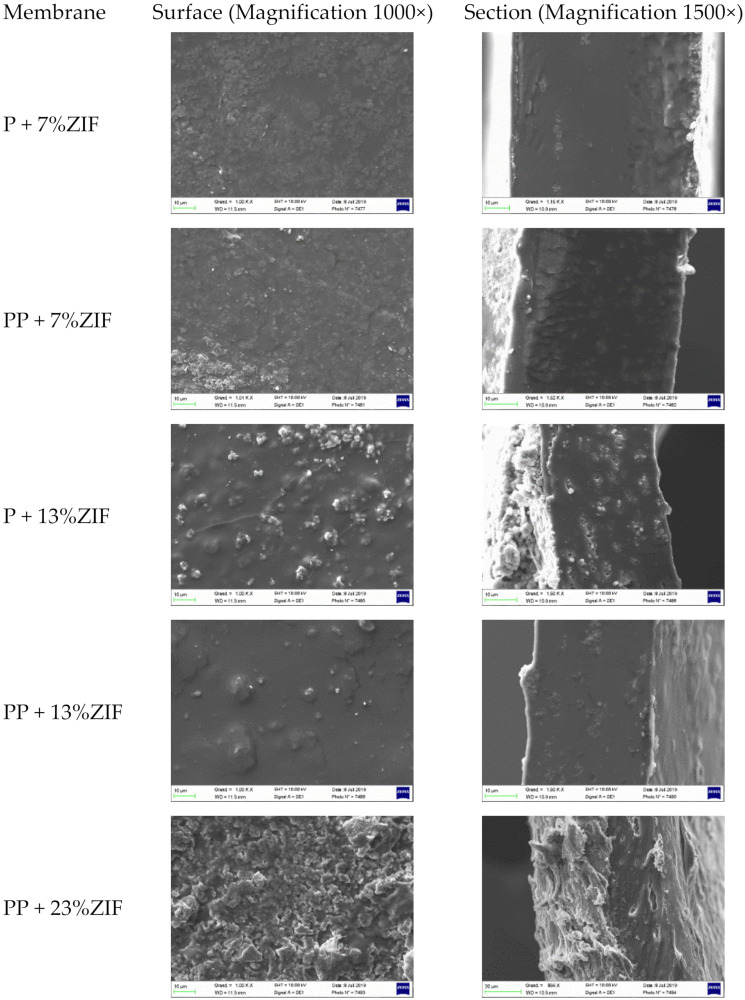
Surface and cross-section morphology of Pebax/ZIF-8 and Pebax/PEG/ZIF-8 membranes.

**Figure 6 membranes-12-00836-f006:**
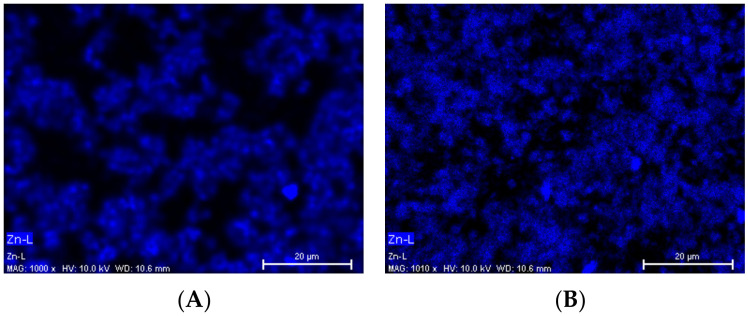
EDX analyses of Pebax/ZIF-8 (**A**) and Pebax/PEG/ZIF-8 (**B**) membranes. ZIF-8 amount is 7 wt%. Blue color is Zn element.

**Figure 7 membranes-12-00836-f007:**
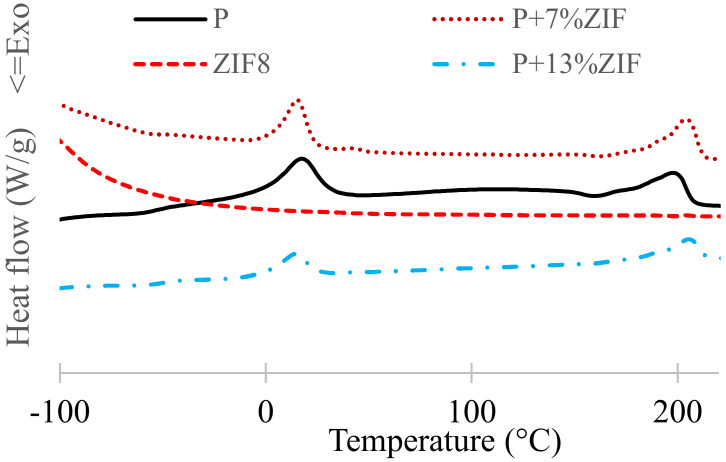
TGA and DSC curves of Pebax/ZIF-8 membranes.

**Figure 8 membranes-12-00836-f008:**
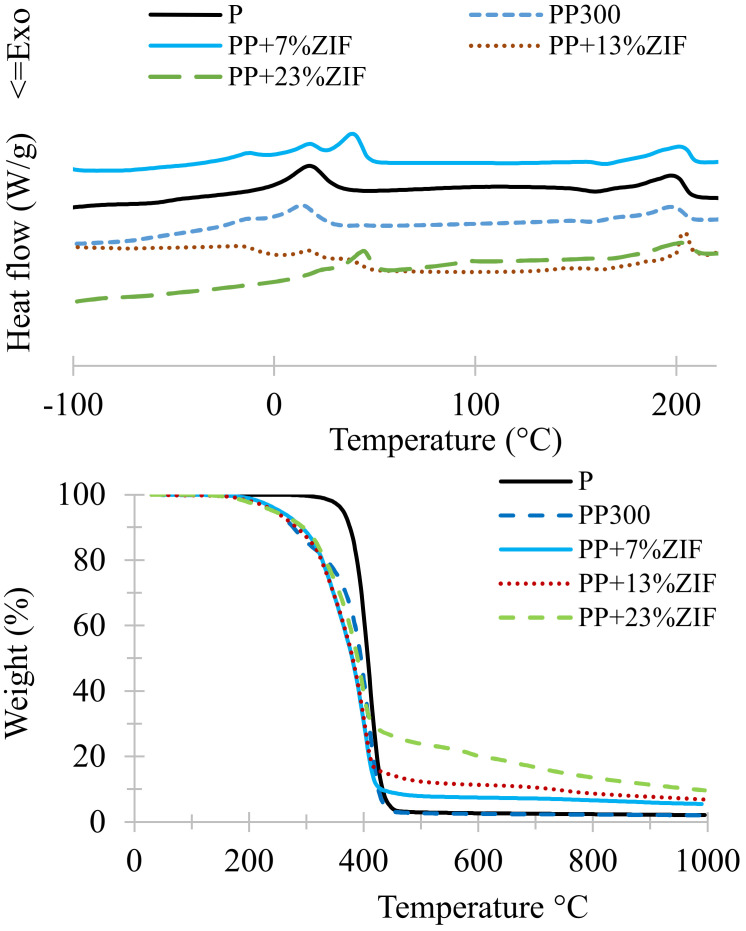
TGA and DSC curves of Pebax/PEG/ZIF-8 membranes.

**Figure 9 membranes-12-00836-f009:**
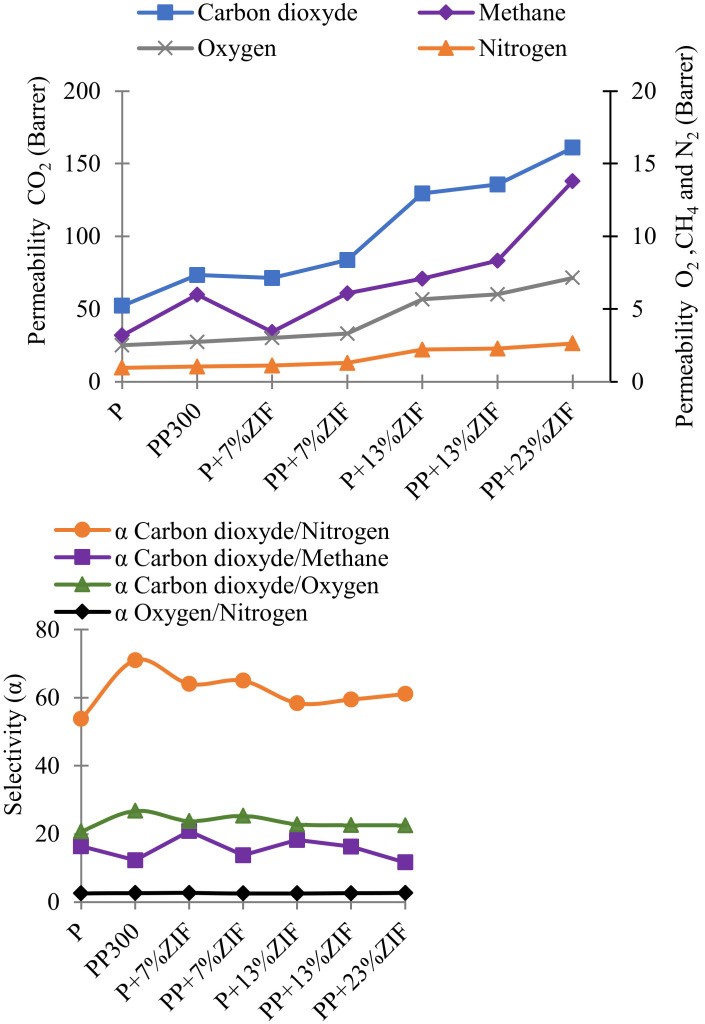
Permeability and selectivity coefficients for neat Pebax, Pebax/ZIF-8 and Pebax/PEG/ZIF-8 blend membranes.

**Figure 10 membranes-12-00836-f010:**
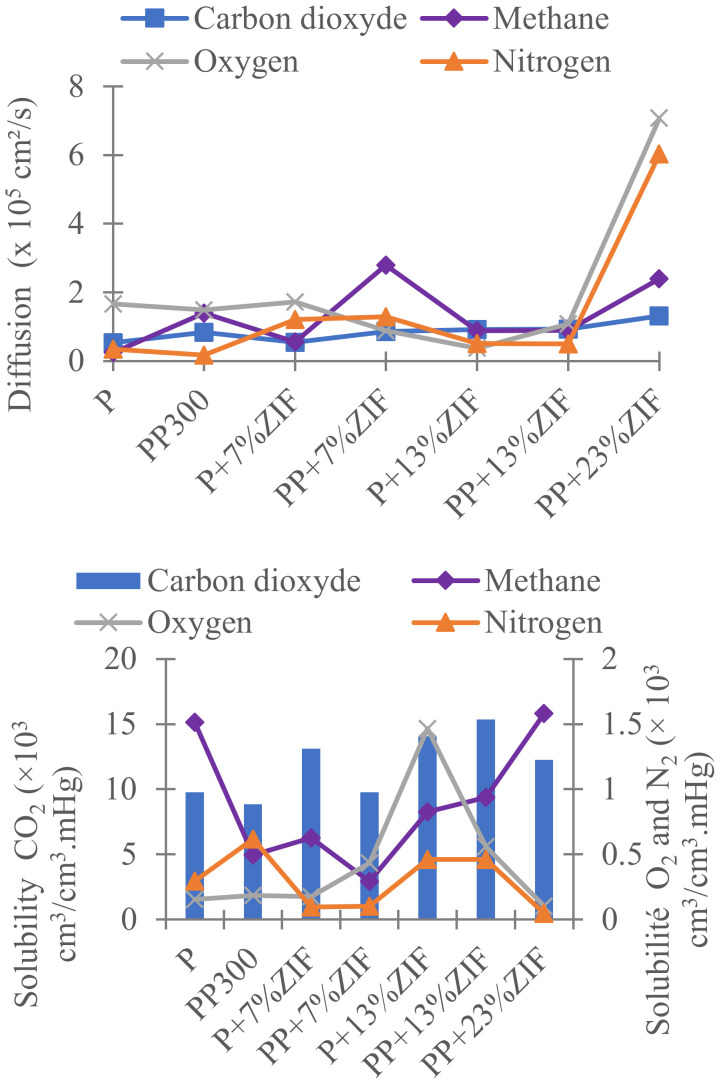
Solubility and diffusion coefficient of CO_2_, CH_4_, N_2_, and O_2_ for neat Pebax, Pebax/ZIF-8 and Pebax/PEG/ZIF-8 blend membranes.

**Figure 11 membranes-12-00836-f011:**
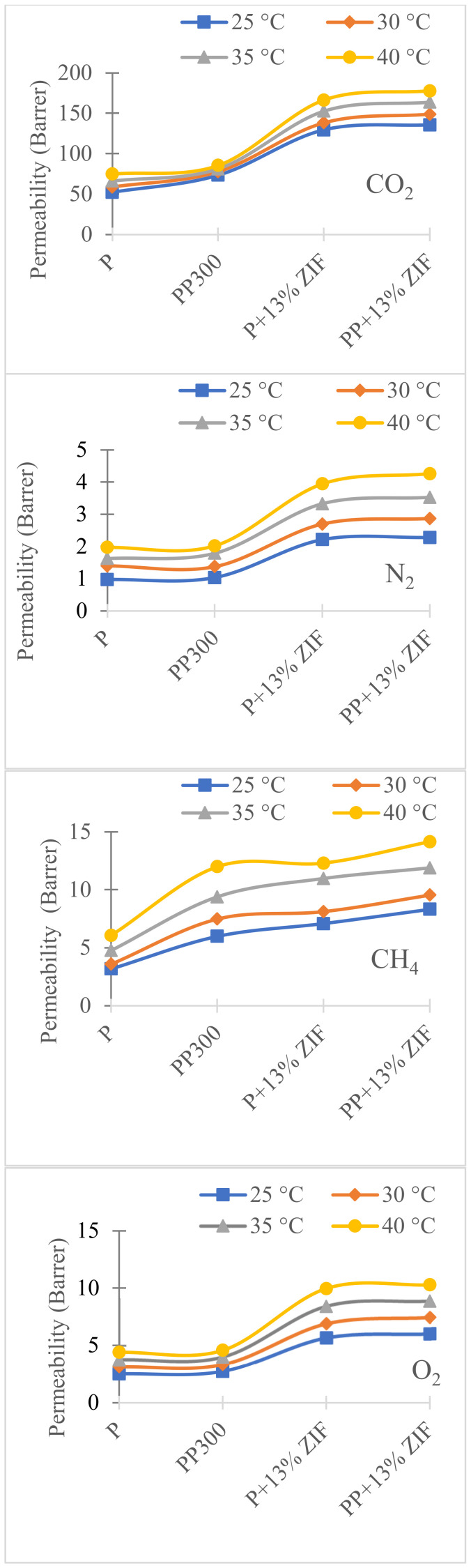
Effect of temperature on permeability for neat Pebax (P), PP300, P + 13% ZIF-8 and PP + 13% ZIF-8.

**Figure 12 membranes-12-00836-f012:**
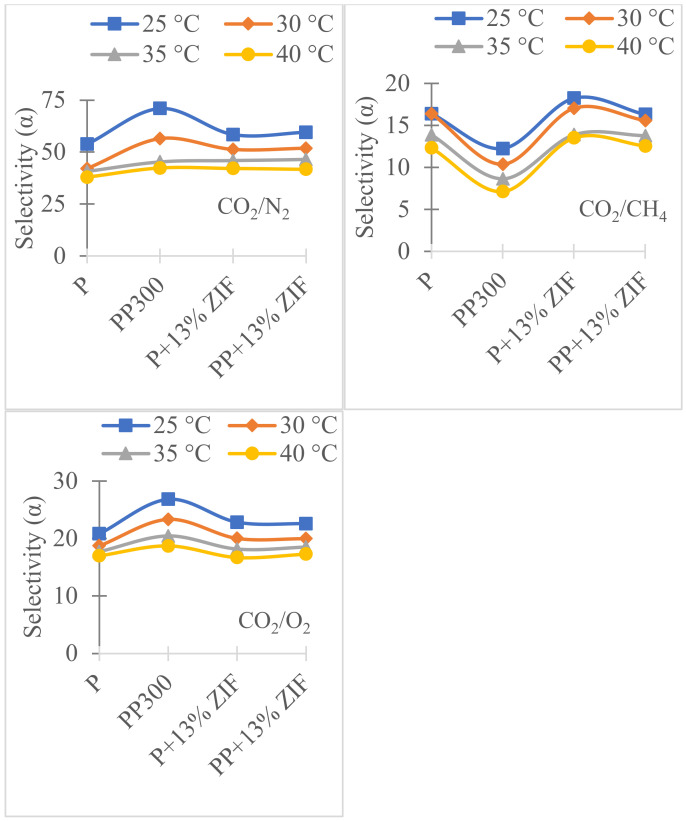
Effect of temperature on selectivity coefficients for neat Pebax (P), PP300, P + 13% ZIF-8 and PP + 13% ZIF-8.

**Figure 13 membranes-12-00836-f013:**
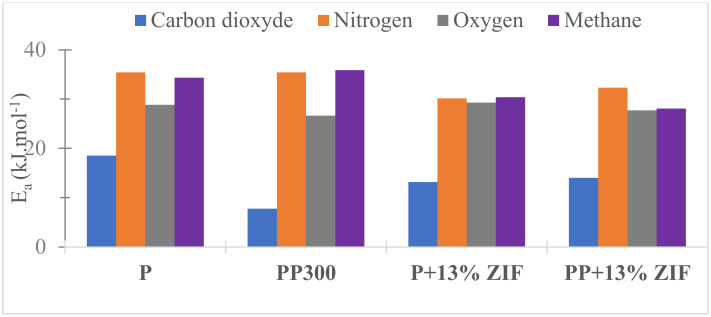
Activation energy of CO_2_, CH_4_, O_2_ and N_2_ for neat Pebax (P), PP300, P + 13% ZIF-8 and PP + 13% ZIF-8.

**Table 1 membranes-12-00836-t001:** Composition of prepared neat Pebax and Pebax-based hybrid membranes.

Membrane	Pebax (wt%)	PEG (wt%)	ZIF-8 (wt%)	ZIF-8/Pebax (wt%)
P	100	-	-	-
PP300	67	33	-	-
PP600	67	33	-	-
PP1500	67	33	-	-
P + 7% ZIF	93	-	7	7
PP + 7% ZIF	64	32	4	7
P + 13% ZIF	87	-	13	13
PP + 13% ZIF	61	30	9	13
PP + 23% ZIF	56	28	16	23

**Table 2 membranes-12-00836-t002:** Thermal properties of neat Pebax and Pebax/PEG membranes.

	TGA				DSC				
Membrane	T1% (°C)	T5% (°C)	DTG (°C)	*R* *(%)*	*T*g (°C)	Tm PEO (°C)	Tm PA (°C)	*Xc* PEO (%)	*Xc* PA (%)
P	327	364	410	2.1	−49	18	198	14	11
PP 300	197	248	405	2.1	−70	14	198	14	7
PP 600	287	330	407	2.8	−61	26	204	34	8
PP 1500	318	359	411	3.2	−50	45	206	49	9

T1%: Temperature for 1% loss of weight decomposition (°C), T5%: Temperature for 5% decomposition (°C), DTG: Temperature of the derivative thermogravimetric peak (°C), *R*: Residual content at 1000 °C (%), *T*g: Glass transition temperature (°C), Tm: Melting temperature (°C); *Xc*: Crystallinity degree (%).

**Table 3 membranes-12-00836-t003:** Thermal properties of neat Pebax, Pebax/ZIF and Pebax/PEG/ZIF membranes.

	TGA				DSC				
Membrane	T1% (°C)	T5% (°C)	DTG (°C)	*R* (%)	*T*g (°C)	Tm PEO (°C)	Tm PA (°C)	*Xc* PEO (%)	*Xc* PA (%)
P	327	364	410	2.1	−49	18	198	14	11
PP 300	197	248	405	2.1	−70	−20–35	198	14	7
P + 7% ZIF	299	333	406	6.2	−47	15	204	11	11
PP + 7% ZIF	194	254	399	5.4	−65	−20–50	201	36	9
P + 13% ZIF	305	334	398	7.5	−47	15	205	8	6
PP + 13% ZIF	179	243	401	6.8	N.D.	−20–50	204	7	6
PP + 23% ZIF	172	242	396	9.6	−58	10–50	202	12	6

T1%: Temperature for 1% loss of weight decomposition (°C), T5%: Temperature for 5% decomposition (°C), DTG: Temperature of the derivative thermogravimetric peak (°C), *R*: Residual content at 1000 °C (%), *T*g: Glass transition temperature (°C), Tm: Melting temperature (°C); ***Xc***: Crystallinity degree (%) N.D: The glass transition temperature is not detected.

## Data Availability

The data presented in this study are available on request from the corresponding author.

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
