# Peer review of "Pebax-Based Composite Membranes with High Transport Properties Enhanced by ZIF-8 for CO2 Separation"

_membranes, 2022, doi:10.3390/membranes12090836_

Round 1

Reviewer 1 Report

Dear Authors,

I studied your manuscript entitled "Pebax-based composite membranes for CO2 separation with high transport properties enhanced by metal organic framework". In my opinion, your paper was written hastily and some spaces need to be improved in terms of journal quality. I recommend major revision before further consideration for publication in the Membranes.

1) From my point of view, the conclusion section should be summarized. It would be helpful if this section contained some more quantitative data.

2) More details about the materials should be provided (especially for Pebax).

3) How do authors select the Pebax/PEG/ZIF ratios? Is there any reference? The authors should spend a few words on this matter. I think that experimental design and optimization techniques should be used to clarify the reasons.

4) Please provide manufacturer details (model, city, or country) for all characterization instruments.

5) In addition to TGA curves, derivative weight curves (DTG) should also be provided.

6) English language needs some polishing (e.g. Typo "Gaz permeation" on page 6.  The article's title is also recommended to be revised.

7) For all Equations, please include the reference for the formula.

Author Response

The authors wish to acknowledge the reviewers and the Editor for their valuable comments and suggestions. We have revised the manuscript following reviewer’s comments. . The authors’ replies to specific comments are provided enclosed.

Best regards 

Reviewer 2 Report

Hello,

In general, the article is tackling the critical and exciting topic of developing a high-performance membrane for CO2 separation. The manuscript is well-structured and well-presented. However, some comments need to be considered to having the article more constructive and of high readership, which is summarized as follows:

1- Careful proofreading is required (hopefully by a  technical English linguist) to have a better representation.

2- I think section 2.3 on membrane performance should be placed last, i.e. after membrane characterizations present in 2.4 and 2.5, this will align as well with the discussion present later in section 3.

3-  In section 2.3 of “Materials and Methods” some results for the accuracy of D, P, and S measurements have been indicated which should be disseminated in the results section.

4- I would suggest providing larger figures for the subfigures in figures 2, 7, and 8 (to be next to each other in vertical orientation rather than horizontal orientation).

5- I suggest providing the data in figures 3, 4, and 9 -12 in tabulated format for easier comparative representation.

Regards

Author Response

(The authors gave the same response as above.)

Round 2

Reviewer 1 Report

Dear Authors,

Thank you for considering my comments. I have recommended the publication of your paper as is.